# Cytotoxicity, Colour Stability and Dimensional Accuracy of 3D Printing Resin with Three Different Photoinitiators

**DOI:** 10.3390/polym14050979

**Published:** 2022-02-28

**Authors:** Gi-Tae Kim, Hye-Bin Go, Jae-Hun Yu, Song-Yi Yang, Kwang-Mahn Kim, Sung-Hwan Choi, Jae-Sung Kwon

**Affiliations:** 1Department and Research Institute of Dental Biomaterials and Bioengineering, Yonsei University College of Dentistry, Seoul 03722, Korea; gitae7673@yuhs.ac (G.-T.K.); hbgo@yuhs.ac (H.-B.G.); syyang88@yuhs.ac (S.-Y.Y.); kmkim@yuhs.ac (K.-M.K.); 2BK21 FOUR Program, Yonsei University College of Dentistry, Seoul 03722, Korea; hun718@yuhs.ac; 3Department of Orthodontics, Institute of Craniofacial Deformity, Yonsei University College of Dentistry, Seoul 03722, Korea

**Keywords:** additive manufacturing, 3D printing accuracy, colour stability, cytotoxicity, DLP 3D printer, TPO-L photoinitiator

## Abstract

Biocompatibility is important for the 3D printing of resins used in medical devices and can be affected by photoinitiators, one of the key additives used in the 3D printing process. The choice of ingredients must be considered, as the toxicity varies depending on the photoinitiator, and unreacted photoinitiator may leach out of the polymerized resin. In this study, the use of ethyl (2,4,6-trimethylbenzoyl) phenylphosphinate (TPO-L) as a photoinitiator for the 3D printing of resin was considered for application in medical device production, where the cytotoxicity, colour stability, dimensional accuracy, degree of conversion, and mechanical/physical properties were evaluated. Along with TPO-L, two conventional photoinitiators, phenylbis (2,4,6-trimethylbenzoyl) phosphine oxide (BAPO) and diphenyl (2,4,6-trimethylbenzoyl) phosphine oxide (TPO), were considered. A total of 0.1 mol% of each photoinitiator was mixed with the resin matrix to prepare a resin mixture for 3D printing. The specimens were printed using a direct light processing (DLP) type 3D printer. The 3D-printed specimens were postprocessed and evaluated for cytotoxicity, colour stability, dimensional accuracy, degree of conversion, and mechanical properties in accordance with international standards and the methods described in previous studies. The TPO-L photoinitiator showed excellent biocompatibility and colour stability and possessed with an acceptable dimensional accuracy for use in the 3D printing of resins. Therefore, the TPO-L photoinitiator can be sufficiently used as a photoinitiator for dental 3D-printed resin.

## 1. Introduction

Traditional dental prostheses have been manufactured in multiple stages. Conventional fabrication methods involve recording an impression of the treatment site, pouring a stone model and constructing a wax pattern. The wax pattern is then invested and replaced with a permanent material such as metal, ceramic, or acrylic resin. The computer-aided design/computer-aided manufacturing (CAD/CAM) system has had a major impact on the dental field by simplifying the process and reducing the production time [1,2]. A 3D printing system that utilizes additive manufacturing has the advantages of being able to create complex shapes, enabling personalized production, and consuming less material than systems using subtractive manufacturing [3,4]. Types of dental 3D printing methods include fused deposition modelling (FDM), material jetting (MJ), selective laser doping (SLS), stereolithography (SLA), and digital light processing (DLP). Among these 3D printing methods, digital light processing (DLP) is a promising technology in dental applications because of its rapid processing, low cost, and high resolution [5]. DLP technology’s method involves the irradiation of light in tanks containing light-cured resin that react with either ultraviolet light or visible light, producing prosthetics through photopolymerization [3]. A digital micromirror device (DMD) is used to cure an entire layer of resin in the XY-axis in one step. The micromirrors, which act as light switches, project the light from the source, as individual pixels, onto the projection surface. Each micromirror represents one or more pixels in the projected image. The number of mirrors corresponds to the resolution of the projected image [5,6]. Dental 3D printing resins can be used to make prostheses such as surgical guides, crowns, bridges, and dentures [7]. The vast majority of dental light-cured resins are composed of dimethacrylate resins [8,9]. The ingredients of a light-cured resin include a matrix, a filler and a photoinitiator to ensure the properties of the material [10]. The resin matrix consists of dimethacrylate monomers such as BisGMA (bisphenol-glycidyl dimethacrylate), UDMA (urethane dimethacrylate), BisEMA (bisphenol-A-ethoxy dimethacrylate), or TEGDMA (triethyleneglycol dimethacrylate). The most commonly used fillers are SiO_2_, TiO_2_, Al_2_O_3_, and ZrO_2_ nanoparticles [11]. A photoinitiator system is added to the resin matrix to trigger the free radical process in the polymerization reaction [12]. The degree of conversion (DC) can be represented as the extent to which monomers react to form polymers. A low degree of conversion has a negative impact on the biological, physical, and mechanical properties of the polymer and ultimately determines the life of the restoration [13,14]. The degree of conversion is affected by several factors, such as the type of methacrylate monomer, temperature, light intensity, and type and concentration of the photoinitiator system [15,16]. Since efficient radical polymerization initiators are a greatly valuable component of light-cured resin, a great deal of research has been conducted to develop the most effective photoinitiator systems [17].

A commonly used photoinitiator in dentistry is camphorquinone (CQ), which is a Norrish type II molecule. It uses a tertiary amine as the main coinitiator and has an absorption wavelength band of 450–500 nm (λmax = 468 nm) [17]. CQ has been frequently used in light-cured resin composites. However, CQ is yellowish, and the amines of the coinitiator undergo yellowing over time, which may impair aesthetics [18]. Furthermore, the polymerization reaction of CQ is slower than that of Norrish type I molecules, and it is difficult to polymerize at a wavelength that is emitted by UV sources, which are used in 3D printers. Due to these limitations, the photoinitiators used in 3D printing systems are primarily Norrish type I photoinitiators [19]. Among Norrish type I phosphines, the phosphine oxide series do not require a coinitiator, unlike CQ, and have the advantages of an excellent degree of conversion, a high rate of polymerization, and stability of colour [20].

Phenylbis (2,4,6-trimethylbenzoyl) phosphine oxide (BAPO) has often been used as a photoinitiator in the 3D printing of resins [19]. BAPO has a high light absorptivity and can divide up to four radicals per molecule, so the polymerization efficiency is excellent [21]; however, it is known to induce discolouration and to be more cytotoxic than TPO and CQ at the same concentration [22]. Diphenyl (2,4,6-trimethylbenzoyl) phosphine oxide (TPO) has been extensively studied as a photoinitiator for polymer-based composites [23,24,25]. TPO can be divided into two radicals, has a higher degree of conversion than CQ, and better colour stability than BAPO [26]; however, the higher cytotoxicity than CQ at the same concentration and a lower polymerization efficiency than BAPO remain problematic [21]. Recently, ethyl (2,4,6-trimethylbenzoyl) phenylphosphinate (TPO-L) has been suggested as a novel photoinitiator that can be used in 3D printing, as Zeng et al., reported that TPO-L showed the highest biocompatibility and excellent transparency among the seven photoinitiators analysed in their study [22], while Steyrer et al., suggested that TPO-L has a higher degree of conversion than BAPO [27].

Nevertheless, limited studies have applied TPO-L to the 3D printing of dental resins to solve the problems associated with BAPO and TPO photoinitiators. Therefore, the purpose of this study was to evaluate the possibility of applying TPO-L as a photoinitiator in a 3D-printed dental polymer and to investigate if such an addition would have resulted in less cytotoxic materials while maintaining the performance related to the 3D printing of dental devices. The null hypothesis was that there would be no difference in cytotoxicity, colour stability, 3D printing accuracy, degree of conversion, or mechanical properties between the 3D printing of dental resins between the three different photoinitiators.

## 2. Materials and Methods

### 2.1. Materials

In this study, bisphenol-A-ethoxy dimethacrylate (Bis-EMA; Sigma–Aldrich, Steinheim, Germany), urethane dimethacrylate (UDMA; Sigma–Aldrich, Steinheim, Germany), and triethyleneglycolglycol dimethacrylate (TEGDMA; Sigma–Aldrich, Steinheim, Germany) were used for 3D printing of the resin matrix. The photoinitiators used in the experiment were phenylbis (2,4,6-trimethylbenzoyl) phosphine oxide (BAPO; IGM Resins, Waalkwijk, The Netherlands), diphenyl (2,4,6-trimethylbenzoyl) phosphine oxide (TPO; IGM Resins, Waalkwijk, The Netherlands), and ethyl (2,4,6-trimethylbenzoyl) phenylphosphinate (TPO-L; IGM Resins, Waalkwijk, The Netherlands). For the 3D-printed green-phase specimen, isopropyl alcohol (IPA; LG Chem Ltd., Yeosu, Korea) was used as the wash solution. The reagents used in the cytotoxicity test were RPMI-1640 cell culture medium (Welgene, Gyeongsan, Korea), foetal bovine serum (FBS; Gibco, Grand Island, NY, USA), penicillin/streptomycin (Gibco, Grand Island, NY, USA), and isopropanol (Sigma-Aldrich, Steinheim, Germany).

### 2.2. Characterization of the Absorption Spectra of Each Photoinitiator

Spectrophotometric analysis was performed to confirm that the absorption spectra of the photoinitiators were consistent with the emission wavelength band of the 3D printer and postcuring unit. Each initiator was completely dissolved in toluene (Sigma-Aldrich, Steinheim, Germany). The concentration of the dissolved photoinitiator was 1 mM. These concentrations were selected to match the mole ratio of the initiators in the materials as tested. Two millilitres of solution was placed in UV-transparent cuvettes, and UV-vis spectra (350–500 nm) were obtained using a UV-vis spectrophotometer (V-650; JASCO, Hachioji, Japan) with a 0.5 nm sampling interval and a 200 nm/min scan speed.

### 2.3. Preparation of 3D-Printing Resin Matrix

The experimental 3D printing resin matrix was formulated using 70 wt% UDMA, 20 wt% Bis-EMA and 10 wt% TEGDMA with a high degree of conversion, according to references [28]. The mixed resin matrix was divided into three groups, according to the three different photoinitiators. The resin mixture in a light-resistant container was added to a photoinitiator at 0.1 mol% and mixed using a magnetic stirrer in a dark environment at 60 °C for 20 min.

### 2.4. Preparation of 3D-Printed Specimens

Software called ‘3D Sprint’ (NextDent Co., Seosterberg, The Netherlands) was used to design the specimen. The specimens required were designed for each. The completed design was positioned in slicing software (VeltzBP; Veltz 3D Co., Incheon, Korea), and after adding a supporter to the specimen design, it was converted to the standard tessellation language (STL) format.

All specimens were manufactured using a DLP printer (D2; Veltz 3D Co., Incheon, Korea). The wavelength was 405 nm, corresponding to visible light, and the accuracy was ± 2 µm. The layer thickness was set to 100 µm, and the light exposure time of the specimen was set to 5.5 s. The 3D-printed green-phase specimen was placed in a beaker containing IPA and ultrasonically cleaned for 3 min. Following that, the supports were removed and ultrasonic cleaning was performed for 2 min using the same method. The support structures were removed using the finish kit (Form 2 Finish Kit; Formlabs Inc., Somerville, MA, USA) and ultrasonic cleaning was performed for 2 min using the same method. After removing all of the remaining IPA with an air gun, specimens were put into a post-curing unit (LC-3D Print Box; NextDent Co., Seosterberg, The Netherlands) with a wavelength band of 350–500 nm and cured for 15 min.

### 2.5. Cytotoxicity Test

To evaluate the cytotoxic effects of each photoinitiator and the 3D-printed resin, a cytotoxicity test was carried out using the 3-(4,5-dimethylthiazol-2-yl)-2,5-diphenyltetrazolium bromide (MTT) assay, according to the ISO 10993-5:2009 standard [29].

Briefly, L-929 mouse fibroblasts were cultured in RPMI-1640 cell culture medium containing 10% foetal bovine serum and 1% penicillin/streptomycin. The cell suspension was prepared at a concentration of 1 × 10^5^ cell/mL and inoculated onto 96-well cell culture plates (100 µL/well). The multi-well plates were incubated at 37 °C with 5% CO_2_ for 24 h. Cells were exposed to either extractions of 3D-printed resin specimens or diluted stock solutions of photoinitiators.

For extraction of the 3D-printed resin specimens, disc-shaped specimens 10 mm in diameter and 5 mm in height were prepared. The residual supports of the 3D-printed specimen were polished with 1000-grit silicon carbide (SiC) paper. Following ISO standard 10993-12:2012 [30], the extract was prepared by soaking specimens in RPMI-1640 culture medium at a concentration of 3 cm^2^/mL. The extraction was carried out at 37 °C for 24 h. The medium of the cultured L929 cells was then replaced by an equal volume (100 µL) of the extract and incubated at 37 °C and 5% CO_2_ in air for 24 h.

In terms of the diluted stock solutions, stock solutions of BAPO, TPO and TPO-L photoinitiators were dissolved to 100 μM in DMSO and freshly diluted in medium prior to each experiment (final DMSO concentration of 1%). The final concentrations of photoinitiators used in MTT assays were between 1 μM and 50 μM. All solutions were prepared under dim light and wrapped in aluminium foil to block light. The medium of the cultured L929 cells was then removed, and L929 cells were cultured in a diluted solution of each photoinitiator at 1 µM, 5 µM, 10 µM, 25 µM, and 50 µM concentrations at 37 °C and 5% CO_2_ for 24 h.

Following removal of the test extracts or diluted stock solutions, 50 µL of MTT solution at a concentration of 1 mg/mL was added to each well and incubated in a dark environment for 2 h at 37 °C. The MTT solution was removed, and 100 µL isopropanol was added to each well. The IPA-treated plate was shaken with a rotator (C-SKS; Changshin Science, Seoul, Korea) for 30 min. The absorbance at 570 nm was measured using a microplate spectrophotometer (Epoch; Biotek, Winooski, VT, USA). The cell viability was calculated using the following equations:(1)$$\mathrm{Viability}\;\%=\frac{100\times OD_{570e}}{OD_{570b}}$$
where *OD*_570_*_e_* is the value of the measured optical density of the extracts of the test sample and *OD*_570*b*_ is the value of the measured optical density of the blank.

### 2.6. Colour Stability

The colour stability after irradiation and water sorption of the 3D printing resin was determined according to ISO 4049:2019 standards [31]. The specimen was disc shaped, 15 mm in diameter and 1.0 mm in height.

The specimens used in the colour stability test were prepared as in Figure 1. The CIELAB coordinates of each specimen were measured using a spectrophotometer (CM-3500d; Konica Minolta, Sensing Inc., Osaka, Japan). In the CIELAB system, the location of a particular shade in the colour space is defined by three coordinates: *L**, *a**, and *b**. *L** respresents the lightness of the object being evaluated. The *a** value represents the colour on the red–green axis and *b** represents the colour valueon the yellow–blue axis. Three sites were measured at random for each specimen, and the mean values and standard deviations were obtained. The classified specimens were subjected to colour comparison (Δ*E*) based on ISO 4049:2019 standards [31]. Δ*E* was calculated using the following equation:(2)$$\Delta E={\left[{\left(\Delta L^{*}\right)}^{2}+{\left(\Delta a^{*}\right)}^{2}+{\left(\Delta b^{*}\right)}^{2}\right]}^{1/2}$$

### 2.7. Evaluation for Dimensional Accuracy

The 3D printing accuracy specimen (*n* = 3) was prepared in the shape of a die, as shown in Figure 2A, based on a reference [32]. The supports were not removed and the specimen was mounted on the plate of the model scanner.

The fabricated specimens were scanned with a light model scanner (Medit T710; Medit, Seoul, Korea) by groups after spraying a scan powder (EASY SCAN SPRAY; Alphadent, Gyeonggi-do, Korea) on the surface of the specimens. The dimensional accuracy between the original STL data and the three different groups was compared by the best-fit alignment using 3D inspection software (Geomagic control X; 3D Systems, Rock Hill, SC, USA). Afterwards, two-dimensional (2D) analysis was performed by dividing the superimposed data equally in the vertical axis. The maximum and minimum range were set at ± 0.5 mm, and the tolerance levels were set at ± 0.05 mm. Each specimen was measured three times, as shown in Figure 2B. The average value was calculated and used.

### 2.8. Degree of Conversion

Specimens (*n* = 3) with a diameter of 10 mm and height of 2 mm were prepared. The prepared disc specimens were polished with 1200-grit silicon carbide (SiC) paper using a water-cooled rotating polishing machine (Ecomet 30; Buehler Ltd., Lake Bluff, IL, USA).

Fourier transform infrared spectroscopy (FT-IR) spectra were recorded using an FT-IR spectrometer (Nicolet iS10, Thermo Scientific, Waltham, MA, USA) with attenuated total reflection (ATR, diamond crystal) accessories. The spectra were obtained in the range of 4000–500 cm^−1^ with a total of 32 scans per spectrum and a resolution of 4 cm^−1^. Three specimens from each experimental group were printed and processed post curing. Each specimen was measured three times with FT-IR, and the average value was calculated and used. Figure 3 shows a comparison of FT-IR spectra before and after conversion.

All the spectra were referenced to the carbonyl group (C=O peak) at 1720 cm^−1^, and the degree of conversion of each specimen was determined by comparing the intensity of the aliphatic C=C stretching vibration at 1638 cm^−1^ of the polymerized 3D printing resin and uncured 3D printing resin. The degree of conversion (*DC*) was determined according to the following equation:(3)$$DC\;\left(\%\right)=\left[1-\frac{\left(1638\;{\mathrm{cm}}^{-1}/1720\;{\mathrm{cm}}^{-1}\right)\mathrm{cured}}{\left(1638\;{\mathrm{cm}}^{-1}/1720\;{\mathrm{cm}}^{-1}\right)\mathrm{uncured}}\right]\times 100$$

### 2.9. Surface Analysis

The Ra value of the disc-shaped specimens (*n* = 3) were measured with an optical three-dimensional surface profilometer (Contour GT-X3 Base, Bruker, Germany). For the surface roughness test, the surface where the layer-by-layer structure was visible was chosen. Three random areas of each specimen were measured at 50.0× magnification in vertical scanning interferometry mode, and the mean Ra value was then calculated.

### 2.10. Three-Point Flexural Strength and Microhardness

The three-point flexural strength specimens (*n* = 15) were referenced to the ISO 4049:2019 standard [31]. The specimens for the three-point flexural strength test was printed with a size of 25 mm × 2 mm × 2 mm. The residual supports of the 3D printed specimens were polished using a water-cooled rotating polishing machine (Ecomet 30; Buehler Ltd., Lake Bluff, IL, USA) with 320-grit silicon carbide (SiC) paper.

A three-point flexural strength test was carried out on a universal testing machine (Instron 5942; Instron, Norwood, MA, USA). Figure 4 shows a stress–strain chart of the average three-point flexural strength of the specimens. The crosshead speed was 1 mm/min and the distance between the two rounded supports was 20 mm. The load was applied until the specimen was fractured. The maximum load was recorded and the flexural strength (*S*) was calculated using the following equation:(4)$$S=\frac{3FL}{2bh^{2}}$$
where *F* is the maximum fracture load, *L* is the distance of the support (20 mm), *b* is the width of the specimen, and *h* is the height of the specimen.

The specimens (*n* = 10) for the microhardness test were disc-shaped, 10 mm in diameter and 5 mm in height. The prepared disc specimens were polished with #400, #800 and #1200 grit silicon carbide (SiC) paper using a water-cooled rotating polishing machine (Ecomet 30; Buehler Ltd., Lake Bluff, IL, USA).

The samples were placed in a Knoop hardness tester (DMH-2; Matsuzawa Seiki Ltd., Tokyo, Japan), and 0.98 N (100 gf) was applied for 25 s. The indentation was observed, and the Knoop hardness number (KHN) was measured to determine the surface hardness. Three sites were measured at random for each specimen, and the mean value and standard deviation were obtained. The hardness was determined using the following equation:(5)$$KHN=14.229Pd^{2}$$
where *P* is the indentation load and *d* is the long diagonal length of the Knoop indentation.

### 2.11. Statistical Analysis

To evaluate the properties of the 3D printing resin with respect to photoinitiators, the results of the cytotoxicity test, the colour stability (Δ*E*), the degree of conversion and the mechanical property data were analysed with a one-way ANOVA followed by Tukey’s statistical test (*p* = 0.05). The dimensional accuracy data were analysed with the Kruskal–Wallis test, followed by the Mann–Whitney post hoc test (*p* = 0.05).

All statistical analyses were performed using the SPSS 25 software program (IBM, Armonk, NY, USA). All statistical significance levels were set at a confidence of 95%.

## 3. Results

### 3.1. Characterization of the Absorption Spectra of Each Photoinitiator

The absorption spectra were obtained from the UV-vis spectrophotometer as shown in Figure 5. The absorption wavelength band of BAPO was 350–450 nm, and TPO and TPO-L showed an absorption wavelength band of approximately 350–430 nm. The emission wavelength band of the 3D printer and postcuring unit was 350–500 nm. Therefore, it was confirmed that the three photoinitiators can sufficiently generate a radical reaction in the emission wavelength band of the 3D printer and the postcuring unit.

### 3.2. Cytotoxicity Test

The results of the cytotoxicity test of the 3D-printed resin using the L-929 mouse fibroblasts are shown in Figure 6, which reveal significant differences among all the groups (*p* < 0.05). The TPO-L group had the highest cell viability (89.62 ± 4.93%), while the BAPO group had the lowest cell viability (74.16 ± 3.7%). The cell viability of the TPO group was 84.45 ± 3.62%. The cell viability of the positive control was 6.77 ± 1.66%.

The results of the cytotoxicity test for each photoinitiator using the L-929 cell line are shown in Table 1. The cell viability of the positive control and negative control was 7.58 ± 1.6% and 95.8 ± 5.3%, respectively. All of the photoinitiators showed no significant differences at concentrations of 1–10 μM; however, there was a significant difference with between concentrations of 25 and 50 μM (*p* < 0.05).

### 3.3. Colour Stability

The colour coordinates according to the conditions of the specimen are shown in Table 2. The BAPO group showed lower values of *a** in the water sorption specimen and lower values of *a** and *b** in the irradiated and non-irradiated specimens compared with the baseline specimen. The TPO and TPO-L groups showed similar values, with lower values of *b** in water sorption, irradiated and non-irradiated specimens, compared with the baseline specimen.

The results of the colour change according to the condition comparisons of the testing groups are shown in Table 3. The BAPO group had the highest Δ*E* values in all the condition comparisons and showed a significant difference from the other groups. The Δ*E* values of the TPO and TPO-L groups showed no significant differences in all condition comparisons (*p* > 0.05). The TPO-L group showed the lowest Δ*E* value in all condition comparisons.

### 3.4. Evaluation for Dimensional Accuracy

The 2D analysis using a colour difference map is shown in Figure 7. The photoinitiator groups are shown in the colour map compared to the original STL file. The TPO and TPO-L groups reveal green lines on the Z-axis and XY-axis. However, the BAPO group reveals a blue line on the Z-axis and a yellow line on the XY-axis.

The 3D printing accuracy results are shown in Table 4. The accuracy of the Z-axis revealed that the TPO-L group was the closest to the true value. The Z-axis value of the TPO group was slightly outside of the tolerance levels but secondarily similar to the true value. The accuracy of the XY-axis indicated that the TPO group was the closest to the true value. The XY-axis value of the TPO-L group was within the tolerance level and secondarily similar to the true value. The BAPO group showed the lowest accuracy on the Z-axis and XY-axis. All groups showed a significant difference (*p* < 0.05). The accuracy of the base plane was not significantly different in all groups (*p* > 0.05).

### 3.5. Degree of Conversion, Surface Analysis and Mechanical Properties

The results of the degree of conversion analysis are shown in Figure 8A. The degree of conversion values were 83.16 ± 3.07% in the BAPO group, 81.75 ± 3.14% in the TPO group, and 85.57 ± 3.87% in the TPO-L group. All groups showed a degree of conversion of more than 80%. The degree of conversion increased gradually from TPO, to BAPO, and then to TPO-L, but there was no significant difference (*p* > 0.05).

The results of the surface analysis are shown in Figure 8B. The Ra values were 1.59 ± 0.29 μm in the BAPO group, 1.61 ± 0.33 μm in the TPO group, and 1.40 ± 0.27 μm in the TPO-L group. The results show no significant differences for each group (*p* > 0.05).

The results of the three-point flexural strength test are shown in Figure 8C. The BAPO group had a strength of 26.18 ± 0.96 MPa, which was not significantly different from 25.75 ± 1.09 MPa for the TPO group and 25.23 ± 1.36 MPa for the TPO-L group (*p* > 0.05).

The results of the Knoop hardness tests are shown in Figure 8D. The hardness values of each group were 141.43 ± 10.38 for the BAPO group, 141.67 ± 14.94 for the TPO group, and 131.1 ± 9.73 for the TPO-L group. The results showed no significant differences for each group in terms of hardness (*p* > 0.05).

## 4. Discussion

Photoinitiators play a key role in the photopolymerization process [33]. The photoinitiation system not only determines the mechanism of the reaction, but also affects the final properties of the polymer, such as the degree of conversion and its mechanical properties. In addition, cytotoxicity and accuracy are very important, as 3D-printed dental resin is placed in the patient’s oral cavity [34]. The selection of an appropriate photoinitiator is essential to obtain the properties of the desired polymer [9,11,17]. A suitable photoinitiator should exhibit a correlation between the absorption characteristics of the photoinitiator and the emission characteristics of the light source. In addition, it should be non-cytotoxic and not cause discolouration [22,35].

In this study, the major aim was to investigate if the addition of TPO-L as a photoinitiator would result in less cytotoxic 3D printing materials compared to the previously used other materials, BAPO and TPO. Additionally, the maintenance of other performance metrics related to 3D printing materials were investigated. The results indicated less cytotoxicity but a comparable performance for 3D printing materials with TPO-L, compared with materials with BAPO or TPO.

Residual monomers and additives are free to diffuse out from the cured materials. They may be released into surrounding tissues and have potentially toxic effects. The photoinitiator was identified as one of the main released components in extracts of resin-based materials [36]. The cytotoxicity of 3D-printed resin groups was tested based on ISO 10993-5:2009 standards [29]. All groups showed significant differences. The group with the lowest cell viability was the BAPO group. According to the references, BAPO shows 50–250-fold higher cytotoxicity than CQ and induces more than 50% cytotoxicity in human oral keratinocyte cells at concentrations greater than 10 µM [20]. In addition, BAPO exhibited a more obvious cytotoxicity response in the HEK293, LO2 and HUVEC cell types [22]. The residual monomers and additives could affect cell viability. Because the degree of conversion of each of the three photoinitiator groups was similar, the amount of residual monomer could be expected to be similar in each photoinitiator group. To evaluate the toxicity of the photoinitiator as an additive, the cytotoxicity of the photoinitiator was measured. The cytotoxicity results of each photoinitiator are shown in Table 1. All photoinitiators showed no significant differences at concentrations of 1–10 μM; however, there was a significant difference at the concentrations of 25 μM and 50 μM. In particular, BAPO and TPO photoinitiators were highly cytotoxic at a concentration of 50 μM. The TPO-L photoinitiator showed the highest cell viability at all concentrations. BAPO and TPO photoinitiators increased cytotoxicity in L929 cells in a concentration-dependent manner. According to the references, TPO reduces cell viability in a dose-dependent manner and TPO is much more cytotoxic than CQ in human pulp-derived cells [21]. Thus, a high concentration of BAPO and TPO is not recommended for clinical application. The TPO-L group exhibited a significantly lower cytotoxicity than the other groups. In other studies, TPO-L was the least toxic of the seven photoinitiators at concentrations of 1–50 μΜ [22]. Therefore, TPO-L is promising for broad application in clinical practice.

The consideration of aesthetics is very important for teeth. For long-term restorations in the oral cavity, the colour of the resin prostheses has a significant impact on patient satisfaction. Therefore, colour stability is an important factor in the success and longevity of restorations [37]. The discolouration of resins can be influenced by a number of factors, such as incomplete polymerization, water sorption, chemical reactivity, diet, oral hygiene, and surface roughness of the restoration [38]. The intrinsic discolouration among the various causes of discolouration mainly depends on the initiator system used in the resins, as well as on the degree of polymerization [39]. As shown by the BAPO group in Table 2, the value of *a** decreased under the water absorption condition and the value of *b** decreased under the irradiation condition, when compared with the baseline condition. Therefore, it was shown that the discolouration of the BAPO group was considerably affected by immersion and light. Additionally, TPO and TPO-L groups found that *b** values decreased and became transparent under the water absorption, irradiation and non-irradiation conditions. Δ*E* values, as shown in Table 3, are important in quantifying the colour difference between two specimens. Under clinical conditions, Δ*E* has to approach 3.3 or higher before the human eye can detect a colour difference [40]. The TPO-L and TPO groups had a Δ*E* value of less than 3.3 in all of the condition comparisons. The TPO-L group presented the lowest Δ*E* value in all of the condition comparisons and showed excellent colour stability. The TPO group had secondarily good colour stability and no significant difference from the Δ*E* value of the TPO-L group (*p* > 0.05). The BAPO group had a Δ*E* value higher than 3.3 for all of the condition comparisons. The cause of the significant colour difference in the BAPO group was the discolouration of the baseline specimen. It can be assumed that the cause of the discolouration was that the acyl radicals remaining after polymerization were generated as coloured radicals [10]. The photoinitiators used in the experiment generate an acyl radical and a phosphonyl radical after the absorption of light energy; however, in the BAPO molecule structure, two carbonyl groups interact with the central phosphonyl group, leading to four reactive radicals [17,41]. As a result, radicals that were not involved in the polymerization were present, and the residual radicals formed coloured radicals, which indicated that the specimen had changed colour. Additionally, the water sorption specimen eluted the components of the immersed specimen, there were no residual radicals, and the colour of the specimens was changed to the colour of the original mixed 3D printing resin. A photoinitiator at a concentration that does not generate the remaining radicals could be expected to avoid or reduce discolouration.

Low accuracy can lead to problems such as the need for chairside adjustment and the compromised longevity of the restoration [42]. The accuracy of the printed objects is correlated with the front polymerization kinetics of formulated resins [43]. The BAPO group showed significantly higher negative errors in the Z-axis than those of the other groups (*p* < 0.05). The formation of a polymer network with a denser structure can result in volumetric contraction or shrinkage [44]. Many factors affect the shrinkage in dental resin. These factors can be separated into material formulation factors (filler content, monomer structure, filler/matrix interactions, additives, etc.) and material polymerization factors (polymerization rate, catalyst and inhibitor concentration, external constraint conditions, curing method, etc.) [45]. As material polymerization factors, photoinitiators affect shrinkage. The BAPO photoinitiator revealed greater shrinkage than from the other photoinitiators. Additionally, the BAPO group showed significantly higher positive errors in the XY-axis when compared to the other groups (*p* < 0.05). These reasons are due to the pixel size of the 3D printer and the high absorbance of the BAPO photoinitiator. The pixel size of the 3D printer used in the experiment was 62.5 μm. When printing the specimen, the BAPO group was overcured by one pixel when compared to the designed value. This resulted in the XY-axis error in the BAPO group. Nevertheless, TPO and TPO-L photoinitiators have less absorbance than BAPO photoinitiators. Therefore, overcuring does not occur in the XY-axis. Current research is, however, limited because there were no experiments with the various 3D printing parameters for shrinkage and over-polymerization. In future research, it will be necessary to change the parameters such as irradiation time, layer thickness, and photoinitiator concentration, as well as to study the optimum concentration of photoinitiators and the resin shrinkage rate.

The degree of conversion is the most important factor in resin materials, as it can affect the mechanical and physical properties as well as the cytotoxicity [28,46,47]. Figure 3 shows that the FT-IR spectra of each group are similar. Therefore, it is expected that the degree of conversion in the experimental groups will be similar. TPO and TPO-L can undergo α-cleavage in the excited triplet state of the C-P bonds after the absorption of light energy, generating two free radicals per molecule, while BAPO forms four radicals per molecule [18]. However, the degree of conversion was not significantly different across all groups (*p* > 0.05). According to Macarie et al., the conversion results in the BAPO group could be explained through the competition between the reactions of primary radicals and double bonds during the initiation stage due to the a higher concentration of radicals [48]. In addition, Lebedevaite et al. confirmed the effect of photoinitiators on photocrosslinking kinetics by the comparison of the G’ curves of the resins. BAPO of 3 mol% demonstrated higher photocrosslinking kinetics compared with photoinitiators of the same concentration. However, the G’ plateau is similar to other photoinitiators [49], which explains why the degree of conversion in the experimental groups is not significantly different.

The quality of the 3D-printed dental restoration is limited by the stair-stepping phenomenon, owing to the layer-by-layer production. It has been found that the use of photoinitiators and photoabsorbers, and the chemical structures thereof, regulate light scattering and penetration, resulting in controlled lateral and vertical printing resolution [50]. Figure 8B shows that the two conventional photoinitiators groups and the TPO-L photoinitiator group have a similar surface roughness (*p* > 0.05).

Appropriate mechanical properties are a necessity for 3D-printed resin because 3D-printed restoration can encounter mechanical stresses when placed on areas that are subjected to mastication force [51]. The mechanical properties of the resin depend on the resin matrix, degree of conversion, and presence of filler [40,52]. In this study, all groups had the same resin matrix mixture. Figure 8C shows that the average flexural strength in all groups was 25–26 MPa. Figure 8D shows that the average Knoop hardness number in all groups was 131–141 KHN. For all groups, we were able to confirm that the mechanical properties were similar and without significant differences. In Figure 8A, all groups showed no significant difference in the degree of conversion. This suggests that the photoinitiators used in the experiment have a similar polymerization effect. The ISO 4049:2019 [31] standard requires a minimum flexural strength of 50 MPa. The reason for the low values from all groups was that no filler was added. Fillers can improve mechanical properties depending on the type and the loading ratio [40,53]. In this research, to remove the variables, a 3D printing resin was used, in which the resin substrate and photoinitiator were mixed without inorganic fillers. However, further studies are needed with the addition of additives such as fillers in order to improve mechanical properties.

The null hypothesis was partially rejected because significant differences were shown depending on the photoinitiator in all experiments, except for the degree of conversion and the mechanical properties.

## 5. Conclusions

An ideal photoinitiator of 3D printing resin should have biocompatibility, colour stability, a high degree of conversion, and 3D-printing accuracy. However, BAPO and TPO photoinitiators, which are mainly used for 3D printing, have disadvantages such as cytotoxicity and discolouration. The purpose of this study was to investigate whether the TPO-L photoinitiator could replace BAPO and TPO photoinitiators.

In this study, the experimental results showed that photoinitiators can significantly affect dental 3D-printed resin. The TPO-L group was either similar or showed better experimental results than the BAPO and TPO groups. In particular, the TPO-L group has the highest cell viability of 89.62 ± 4.93% in the cytotoxicity test of 3D-printed resin. In the colour stability test, a Δ*E* value of 0.75 to 1.54 was shown in all condition comparisons and showed excellent colour stability. In terms of dimensional accuracy, the TPO-L group had the least error in the value of −0.007 ± 0.005 mm on the Z-axis, and the XY-axis had the value of −0.048 ± 0.021 mm, which did not exceed the tolerance levels. Therefore, the TPO-L photoinitiator showed excellent biocompatibility, colour stability, and sufficient accuracy for use in 3D printing. In conclusion, the TPO-L photoinitiator can solve the problems associated with BAPO and TPO photoinitiators and be used as a photoinitiator for the 3D printing of dental resin.

## Figures and Tables

**Figure 1 polymers-14-00979-f001:**
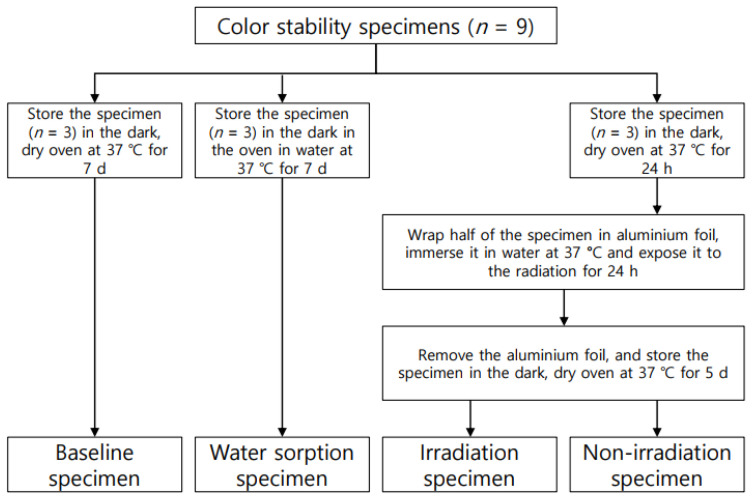
Classification of specimens for colour stability test.

**Figure 2 polymers-14-00979-f002:**
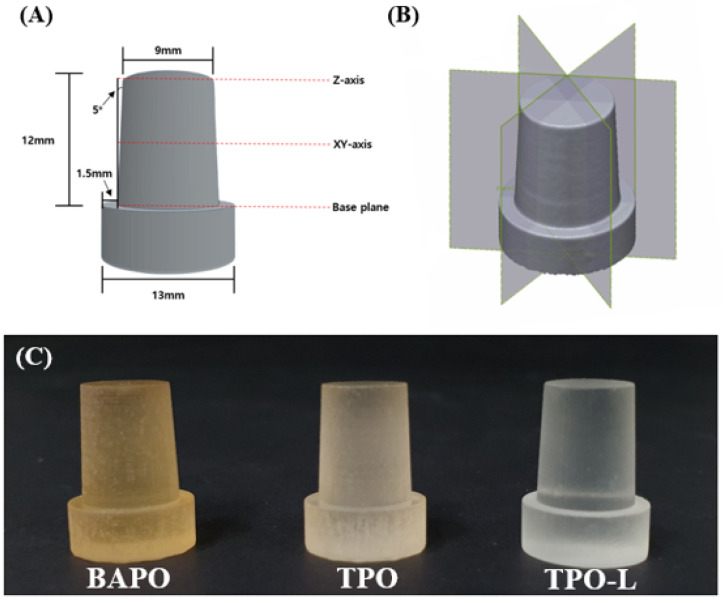
(**A**) Specimen design for the 3D printing accuracy test (**B**) plane division for 2D analysis, and (**C**) 3D-printed accuracy specimens.

**Figure 3 polymers-14-00979-f003:**
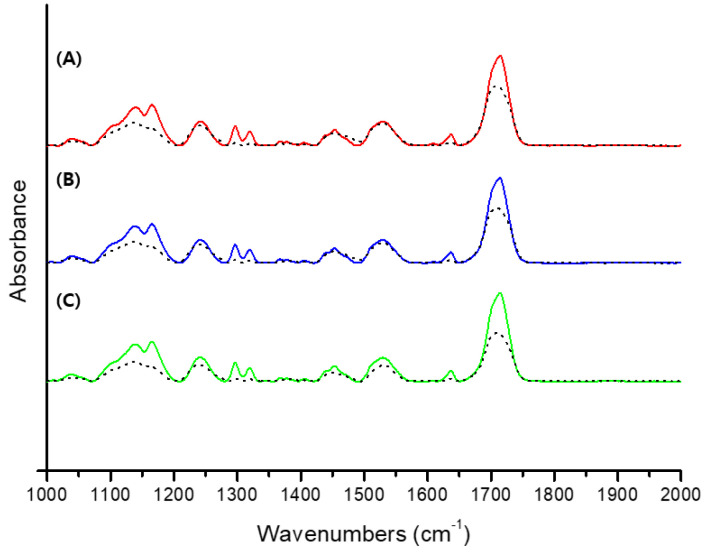
Comparison of FT-IR spectra before and after conversion. (**A**) Comparison of the BAPO group, (**B**) comparison of the TPO group, (**C**) comparison of the TPO-L group. The black dashed line shows the after conversion.

**Figure 4 polymers-14-00979-f004:**
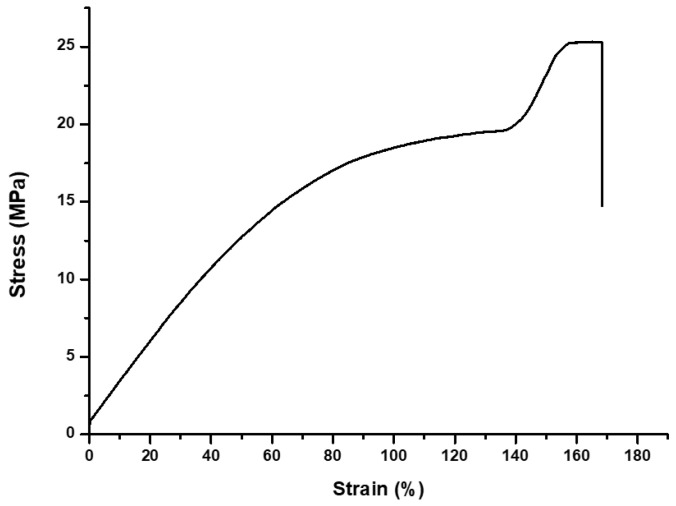
Average stress–strain chart of the experimental groups.

**Figure 5 polymers-14-00979-f005:**
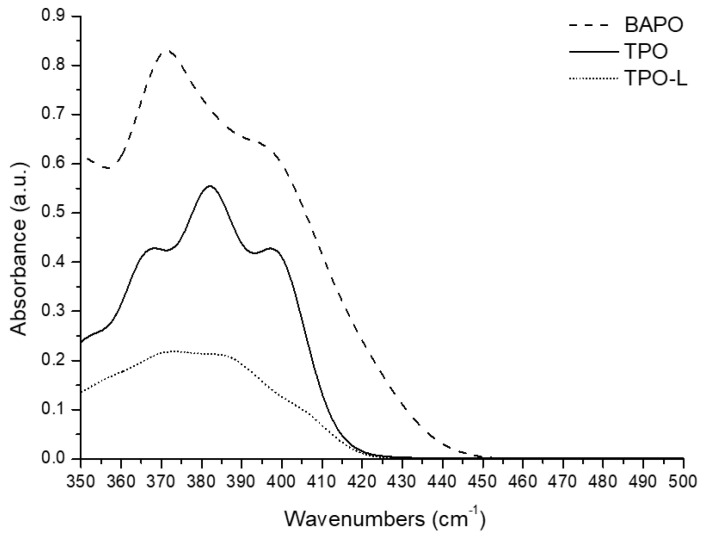
Absorption spectra for each photoinitiator. BAPO—phenylbis (2,4,6-trimethylbenzoyl) phosphine oxide; TPO—diphenyl (2,4,6-trimethylbenzoyl) phosphine oxide; TPO-L—ethyl (2,4,6-trimethylbenzoyl) phenylphosphinate.

**Figure 6 polymers-14-00979-f006:**
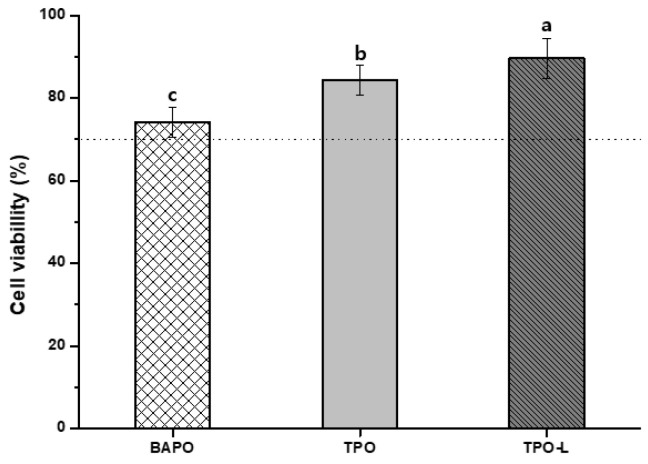
Cell viability of 3D-printed resin groups and each photoinitiator. Differences in lowercase alphabetical letters above the bar graph indicate significant differences of each group (*p* < 0.05).

**Figure 7 polymers-14-00979-f007:**
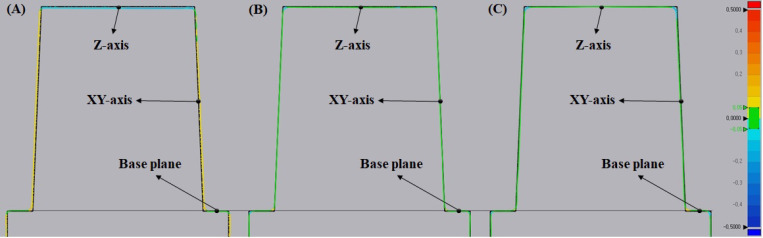
Two-dimensional analysis of accuracy using a colour difference map (green represents a good fit, yellow or red represents a positive error, and blue represents a negative error). (**A**) Two-dimensional analysis of the BAPO group, (**B**) two-dimensional analysis of the TPO group, (**C**) two-dimensional analysis of the TPO-L group. The black dashed line shows the base plane.

**Figure 8 polymers-14-00979-f008:**
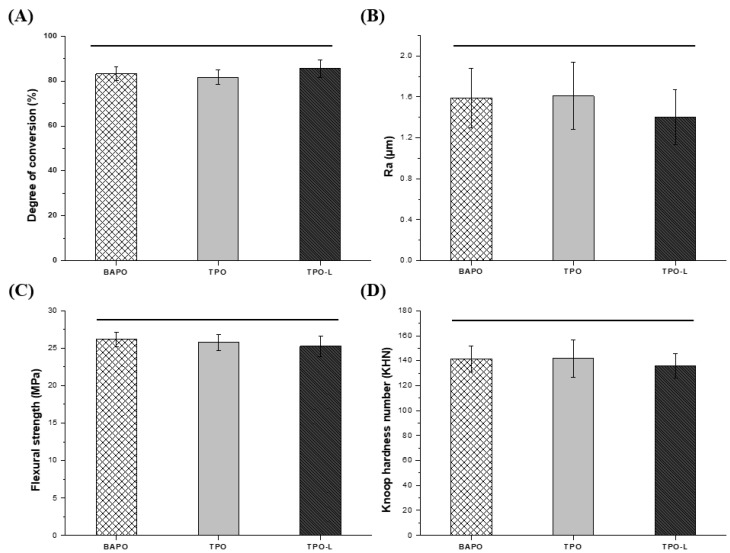
(**A**) Degree of conversion of each group. (**B**) Surface analysis of each group. Mechanical properties in terms of (**C**) flexural strength and (**D**) microhardness of each group. Horizontal bars indicate no significant difference from each other (*p* > 0.05).

**Table 1 polymers-14-00979-t001:** Cell viability of each photoinitiator (%).

Testing Groups	1 μM	5 μM	10 μM	25 μM	50 μM
BAPO	92.98 ± 6.93 ^a^	83.69 ± 9.69 ^a^	83.12 ± 10.11 ^a^	70.44 ± 11.84 ^b^	56.35 ± 10.40 ^b^
TPO	93.35 ± 8.08 ^a^	92.01 ± 10.21 ^a^	85.14 ± 11.94 ^a^	76.80 ± 9.36 ^ab^	61.84 ± 15.60 ^ab^
TPO-L	95.91 ± 4.85 ^a^	95.48 ± 5.91 ^a^	95.57 ± 6.37 ^a^	91.85 ± 8.85 ^a^	80.48 ± 8.46 ^a^

The same lowercase letter in the same column indicates that there is no significant difference (*p* > 0.05).

**Table 2 polymers-14-00979-t002:** Change in colour coordinates.

Testing Groups	Baseline			Water Sorption			Irradiation			Non-Irradiation		
	*L**	*a**	*b**	*L**	*a**	*b**	*L**	*a**	*b**	*L**	*a**	*b**
BAPO	36.2	1.21	7.79	39.63	−3.6	7.65	33.75	−0.53	1.52	36.9	−2.42	5.48
TPO	31.75	−0.23	1.82	32.01	−0.41	0.49	33.76	−0.39	0.74	33.76	−0.35	0.64
TPO-L	31.96	−0.28	1.40	32.31	−0.38	0.48	32.07	−0.34	0.52	31.47	−0.29	0.55

**Table 3 polymers-14-00979-t003:** Colour change according to condition comparison (Δ*E*).

Testing Groups	Baseline and Water Sorption	Baseline and Irradiation	Baseline and Non-Irradiation	Irradiation and Non-Irradiation
BAPO	5.59 ± 1.43 ^a^	7.21 ± 1.1 ^a^	5.26 ± 1.15 ^a^	5.65 ± 0.27 ^a^
TPO	1.79 ± 0.35 ^b^	2.73 ± 1.38 ^b^	2.45 ± 1.34 ^b^	2.59 ± 1.22 ^b^
TPO-L	1.54 ± 0.61 ^b^	1.24 ± 0.27 ^b^	1.05 ± 1.43 ^b^	0.75 ± 0.7 ^b^

The same lowercase letter in the same column indicates that there is no significant difference (*p* > 0.05).

**Table 4 polymers-14-00979-t004:** Mean and standard deviation of 3D printing accuracy according to three photoinitiators.

Testing Groups	Z-Axis (mm)	XY-Axis (mm)	Base Plane (mm)
BAPO	−0.102 ± 0.037 ^c^	0.06 ± 0.030 ^c^	0.013 ± 0.018 ^a^
TPO	−0.053 ± 0.003 ^b^	−0.021 ± 0.013 ^a^	0.018 ± 0.017 ^a^
TPO-L	−0.007 ± 0.005 ^a^	−0.048 ± 0.021 ^b^	0.028 ± 0.018 ^a^

The same lowercase letter in the same column indicates no significant difference (*p* > 0.05).

## Data Availability

Not applicable.

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
