# Peer review of "Cytotoxicity, Colour Stability and Dimensional Accuracy of 3D Printing Resin with Three Different Photoinitiators"

_polymers, 2022, doi:10.3390/polym14050979_

Round 1
Reviewer 1 Report
The manuscript describes research related to the new type of resin dedicated for use in additive manufacturing. The abstract, which is well written indicates all activities from the chemical analysis of the produced resin to mechanical properties of 3D printed parts.
The introduction part is very well written with all necessary explanations of the influence of each chemical element on polymerization and additive manufactured part properties. Each part of the introduction is properly supported by citations.
In the research description, all necessary parts of the authors' research are properly designed and described in the manuscript. During the revision, I noticed two issues that should be taken into account during the manuscript correction:
- There is no explanation of the used p<0.05 in the text.
- In the mechanical properties part, there should be a stress-strain chart to better understand the material behavior during three-point flexural strength analysis.
- There should be some quantified values in the conclusion part.
After the correction of those two minor issues, the manuscript will be ready for publication.
Author Response
Reviewer #1 |
The manuscript describes research related to the new type of resin dedicated for use in additive manufacturing. The abstract, which is well written indicates all activities from the chemical analysis of the produced resin to mechanical properties of 3D printed parts. The introduction part is very well written with all necessary explanations of the influence of each chemical element on polymerization and additive manufactured part properties. Each part of the introduction is properly supported by citations. In the research description, all necessary parts of the authors' research are properly designed and described in the manuscript. During the revision, I noticed two issues that should be taken into account during the manuscript correction: |
Comment 1 |
There is no explanation of the used p<0.05 in the text. |
Response to Comment 1 |
Thank you for the valuable suggestions. The relevant information has been added according to your comments as follows: 2. Materials and Methods 2.10. Statistical analysis To evaluate the properties of the 3D printing resin according to photoinitiators, the results of the cytotoxicity test, colour stability (ΔE), degree of conversion and mechanical property data were analysed with one-way ANOVA followed by Tukey’s statistical test (p=0.05). The results of dimensional accuracy data were analysed with the Kruskal–Wallis test followed by the Mann–Whitney post hoc test (p=0.05). |
Comment 2 |
In the mechanical properties part, there should be a stress-strain chart to better understand the material behavior during three-point flexural strength analysis. |
Response to Comment 2 |
According to your suggestion, the related sentences and figure were added in the methods sections. 2. Materials and Methods 2.10. Three-point flexural strength and microhardness ~ Figure 3 shown a stress-strain chart of average three-point flexural strength of specimens. ~
Figure 3. Average stress-strain chart of the experimental groups. |
Comment 3 |
There should be some quantified values in the conclusion part. |
Response to Comment 3 |
According to your suggestion, we have revised the conclusion part as follows: 5. Conclusions ~ In particular, the TPO-L group has the highest cell viability of 89.62 ± 4.93 % in the cytotoxicity test of 3D printed resin. In the colour stability test, the ΔE value of 0.75 to 1.54 was shown in all condition comparisons and showed excellent colour stability. In terms of dimensional accuracy, the TPO-L group had the least error in the value of (-0.007 ± 0.005) mm on the Z-axis, and the XY-axis had the value of (-0.048 ± 0.021) mm, which did not exceed the tolerance levels. Therefore, ~ |
Reviewer 2 Report
The authors present a simple manuscript. The results are simple and in some doubtful cases, the figures do not allow conclusions to be taken as categorical as those presented by the authors. The discussion is vague and does not have a translational and innovative character.
This manuscript is not suitable for publication in a Q1 journal.
Author Response
Reviewer #2 |
The authors present a simple manuscript. The results are simple and in some doubtful cases, the figures do not allow conclusions to be taken as categorical as those presented by the authors. The discussion is vague and does not have a translational and innovative character. This manuscript is not suitable for publication in a Q1 journal. |
Response to Comment |
Thank you for your valuable advice on the manuscript. First, we have thoroughly revised the manuscript with details on scientific novelty of this paper along with modifications and additions of figures that would link to conclusion. Also, we revised discussion section with clear link of findings. Finally, we proved the availability of TPO-L photoinitiator with image of the printed object and surface analysis. Hopefully, this may have improved the manuscript. |
Reviewer 3 Report
Dear Editor,
The MS is focused on the use of ethyl (2,4,6-trimethylbenzoyl) phe-nylphosphinate as a photoinitiator of 3D printing resin was considered for application in medical device production, where the cytotoxicity, color stability, dimensional accuracy, degree of conversion, mechanical and physical properties were evaluated. There are some comments on this work.
- There is no image of the printed object.
- There should be surface analysis data for printed objects.
- The introduction should be extended a little bit.
- The discussion of cytotoxicity assays are not enough
Author Response
Reviewer #3 |
The MS is focused on the use of ethyl (2,4,6-trimethylbenzoyl) phenylphosphinate as a photoinitiator of 3D printing resin was considered for application in medical device production, where the cytotoxicity, color stability, dimensional accuracy, degree of conversion, mechanical and physical properties were evaluated. There are some comments on this work. |
Comment 1 |
There is no image of the printed object. |
Response to Comment 1 |
Thank you for the valuable suggestions. The image of the printed object was attached according to the customer's suggestion. 2. Materials and Methods 2.7. Evaluation for dimensional accuracy Figure 2. Specimen design for 3D printing accuracy test (A), plane division for 2D analysis (B) and 3D printed accuracy specimens (C). |
Comment 2 |
There should be surface analysis data for printed objects. |
Response to Comment 2 |
According to your suggestion, The relevant information has been added according to your comments as follows: 2. Materials and Methods 2.9. Surface analysis The Ra value of disk-shaped specimen (n=3) was measured with an optical 3-dimensional surface profilometer (Contour GT-X3 Base, Bruker, Germany). For the surface roughness test, the surface where layer by layer structure was visible was chosen. Three random areas of each specimen were measured at 50.0× magnifications in vertical scanning interferometry mode, and the mean Ra value was then calculated. 3. Results 3.5. Degree of conversion, surface analysis and mechanical properties ~ The results of the surface analysis are shown in Figure 7 (B). The Ra values were 1.59 ± 0.29 μm in the BAPO group, 1.61 ± 0.33 μm in the TPO group, and 1.40 ± 0.27 μm in the TPO-L group. The results showed no significant differences for each group (p>0.05). ~
Figure 7. (A) Degree of conversion of each group. Horizontal bars indicate no significant difference from each other (p>0.05). (B) Surface analysis of each group. Horizontal bars indicate no significant difference from each other (p>0.05). Mechanical properties in terms of (C) flexural strength and (D) microhardness of each group. Horizontal bar: The microhardness of each group was not significantly different from each other (p>0.05). 4. Discussion ~ The quality of the dental 3D printed restoration is limited by a stair-stepping phenomenon owing to the layer-by-layer production. It has been found that the use of photoinitiators and photoabsorbers, or the chemical structures thereof, regulated light scattering and penetration, resulting in controlled lateral and vertical printing resolution [46]. Figure 7 (B) shows that the two conventional photoinitiators groups and the TPO-L photoinitiator group have a similar surface roughness (p>0.05). ~
|
Comment 3 |
The introduction should be extended a little bit. |
Response to Comment 3 |
According to your suggestion, we have revised the manuscript as follows: 1. Introduction ~ The CAM system is divided into additive manufacturing and subtractive manufacturing. A 3D printing system that corresponds to additive manufacturing has the advantage of being able to create complex shapes, personalized production and consuming less material [3,4]. Dental 3D printing resins can be used to make prostheses such as surgical guides, crowns & bridges and dentures [5]. Types of dental 3D printing methods include fused deposition modeling (FDM), Material jetting (MJ), selective laser doping (SLS), stereolithography (SLA) and digital light processing (DLP). Among the 3D printing methods, digital light processing (DLP) is a promising 3D printing technology in dental applications because of its rapid processing, low cost and high resolution [6]. DLP technology is the method of irradiating light in tanks containing light-cured resin that reacts with ultraviolet light and produces prosthetics through photopolymerization [3]. When light irradiation, a digital micromirror device (DMD) is used to cure the entire layer of resin in the x-y axis at one time. The DMD consists of thousands of movable micro-mirrors, and the higher the number of micromirrors, the higher the resolution. ~ |
Reviewer 4 Report
In this manuscript the authors studied the use of BAPO, TPO and TPO-L as photoinitiator for 3D printing applications. The authors examined the influenced of PI on cytotoxicity, colour stability, dimensional accuracy, degree of conversion and mechanical properties. The TPO-L photoinitiator showed excellent biocompatibility and colour 28 stability and possessed acceptable dimensional accuracy for use in 3D printing resins. The work looks interesting and provided a potential to serve as a photoinitiator for 3D printing dental resin. By considering the literature survey and data information provided, the quality of this manuscript is suitable to this journal. So I recommend for acceptation of this manuscript. My comments are listed as below.
1) Page 3, Line 128. (IPA; LG Chem Ltd., Yeosu, Korea). I suggest authors to move this information to Section 2.1 Materials.
2) Page 3, Line 140 to 144, 165. I suggest authors to move the chemical descriptions part to Section 2.1 also.
3) Page 4, Line 180. Please include the definition for L, a, b.
4) Page 6, Line 214. Please give the FTIR spectra comparison of before vs. after conversion.
5) Page 8, Line 278 to 287. I suggest authors to summarize the cytotoxicity test data in table format. It will be helpful from reader point of view.
6) Please comment on the curing kinetics for the 3 PI used (BAPO, TPO, TPO-L). I suggest authors to put an overlay conversion curve to compare the kinetics of the 3 PI under same molar concentration.
Author Response
Reviewer #4 |
In this manuscript the authors studied the use of BAPO, TPO and TPO-L as photoinitiator for 3D printing applications. The authors examined the influenced of PI on cytotoxicity, colour stability, dimensional accuracy, degree of conversion and mechanical properties. The TPO-L photoinitiator showed excellent biocompatibility and colour stability and possessed acceptable dimensional accuracy for use in 3D printing resins. The work looks interesting and provided a potential to serve as a photoinitiator for 3D printing dental resin. By considering the literature survey and data information provided, the quality of this manuscript is suitable to this journal. So I recommend for acceptation of this manuscript. My comments are listed as below. |
||||||||||||||||||||||||
Comment |
1) Page 3, Line 128. (IPA; LG Chem Ltd., Yeosu, Korea). I suggest authors to move this information to Section 2.1 Materials. 2)Page 3, Line 140 to 144, 165. I suggest authors to move the chemical descriptions part to Section 2.1 also. |
||||||||||||||||||||||||
Response to Comment |
Thank you for your thoughtful comments on the manuscript. We have moved the relevant information to Section 2.1 Materials. 2. Materials and Methods 2.1. Materials ~ For the 3D-printed green phase specimen, isopropyl alcohol (IPA; LG Chem Ltd., Yeosu, Korea) was used as the wash solution. The reagents used in the Cytotoxicity test were RPMI-1640 cell culture medium (Welgene, Gyeongsangbuk-do, Korea), fetal bovine serum (FBS; Gibco, Grand Island, NY, USA), penicillin/streptomycin (Gibco, Grand Island, NY, USA) and isopropanol (Sigma–Aldrich, Steinheim, Germany). |
||||||||||||||||||||||||
Comment |
3) Page 4, Line 180. Please include the definition for L, a, b. |
||||||||||||||||||||||||
Response to Comment |
The relevant information has been added according to your comments as follows: 2. Materials and Methods 2.6. Colour stability ~ In the CIELAB system, the location of a particular shade in the colour space is defined by three coordinates: L*, a*, and b*. L* describes the lightness of the object being evaluated. The a* value defines the colour on the red-green axis and b* on the yellow-blue axis. ~ |
||||||||||||||||||||||||
Comment |
4) Page 6, Line 214. Please give the FTIR spectra comparison of before vs. after conversion. |
||||||||||||||||||||||||
Response to Comment |
According to your suggestion, we have added a FT-IR spectra in following section; 2. Materials and Methods 2.8. Degree of conversion Figure 3. Comparison of before and after conversion of FT-IR spectra. (A) Comparison of the BAPO group, (B) comparison of the TPO group, (C) comparison of the TPO-L group. The black dashed line shows the after conversion. |
||||||||||||||||||||||||
Comment |
5) Page 8, Line 278 to 287. I suggest authors to summarize the cytotoxicity test data in table format. It will be helpful from reader point of view. |
||||||||||||||||||||||||
Response to Comment |
According to your suggestion, we have revised the cytotoxicity test part in following section: 3. Results 3.2. Cytotoxicity test ~ The results of the cytotoxicity test of each photoinitiator are shown in Table 1. The cell viability of the positive control and negative control was 7.58 ± 1.6% and 95.8 ± 5.3%, respectively. All photoinitiators showed no significant differences at concentrations of 1~10 μM. However, there was a significant difference in the concentrations of 25 μM and 50 μM (p<0.05).
Figure 5. Cell viability of 3D printed resin groups. Differences in lowercase alphabetical letters above the bar graph indicate significant differences of each group (p<0.05). Table 1. Cell viability of each photoinitiator (%).
The same lowercase letter in the same column indicates no significant difference (p>0.05). |
||||||||||||||||||||||||
Comment |
6) Please comment on the curing kinetics for the 3 PI used (BAPO, TPO, TPO-L). I suggest authors to put an overlay conversion curve to compare the kinetics of the 3 PI under same molar concentration. |
||||||||||||||||||||||||
Response to Comment |
Thank you for the valuable suggestions. we modified the discussion part by adding references that measured the curing kinetics by real-time photorheometry of the 3 PI under the same molar concentration. 4. Discussion ~ Figure 3 shows that the FT-IR spectra of each group are similar. Therefore, it could be expected that the degree of conversion of the experimental groups will be similar. ~ According to Macarie et al., conversion results of BAPO group could be explained by the competition between the reactions of primary radicals and double bonds during the initiation stage due to the higher concentration of radicals [48]. In addition, Lebedevaiteet al. confirmed the effect of photoinitiators on photocrosslinking kinetics by comparison of the G’ curves of the resins. 3 mol% BAPO demonstrated higher photocrosslinking kinetics compared to photoinitiators of the same concentration. However the G' plateau is similar to other photoinitiators [49]. That explains why the degree of conversion in the experimental groups is not significantly different. 48. Macarie, L.; Ilia, G. The influence of temperature and photoinitiator concentration on photoinitiated polymerization of diacrylate monomer. Central European Journal of Chemistry 2005, 3, 721-730. 49. Lebedevaite, M.; Ostrauskaite, J. Influence of photoinitiator and temperature on photocross-linking kinetics of acrylated epoxidized soybean oil and properties of the resulting polymers. Industrial Crops and Products 2021, 161, 113210. |
Reviewer 5 Report
Comments:
This paper titled: Biocompatible 3D printing resin for medical devices using ethyl (2,4,6-trimethylbenzoyl) phenylphosphinate as a photoinitiator presents a study using 3D printable resins using three different photoinitiators: BAPO, TPO and TPO-L. The objective is to show the better biocompatibility, color stability and printing accuracy of TPO-L compared to the other two types of photoinitiators. The TPO-L photoinitiator is a novel compound that can be efficiently used for 3D printing applications, however, in this manuscript. Some changes need to be made to increase the clarity of the investigation and scientific methodology.
The overall English language is acceptable, with some changes throughout the paper. Sometimes, the sense of the sentences is unclear to understand.
The beginning of the introduction is a bit dispersive, and the sense of the work might be lost.
In general, the work is very interesting, and with a little effort, the work can be enhanced and easier to understand the methodology.
Line 35:
Which stages are the authors talking about? Can the authors provide a few examples to enrich this part?
Line 37: From a personal view, I find the CAM system a bit old-fashioned term. Since this paper is talking almost uniquely about 3D printing, I recommend authors use a more-updated term for referring to these technologies
Line 41: consuming less material compared to what?
Line 41: I suggest authors shift this part of the paragraph since 3D printable resins are introduced in the subsequent section.
Line 47: Authors should be careful of using indistinct methods and technology when discussing 3D printing. The same for light-curable and light-cured resins, there is a huge difference between these two terms.
At last, DLP-based 3D printing methods also works with visible light and not only with UV.
Indeed, as the same authors explain later, they use a visible-light 3D printer and not a UV-based one.
line 50: For the sake of printing resolution, authors should mention what each of the micromirror devices represents in terms of image resolution to connect to better explain why a "higher number of micromirrors, the higher is the resolution"
Line 53: Authors should provide some examples and/or references in this part of the paragraph.
Line 54: what do the authors refer to matrix and fillers? Some example?
Line 55: I suggest changing line 55. It is not clear what the purpose of the photoinitiator is: is it used to start the free radical polymerization, complete the polymerization, or both?
Line 56: What authors refer to methacrylate-based material? A solid or liquid material? Why did the authors start to talk about methacrylate-based materials without being previously introduced?
Typically, for light-based 3D printing, meth(acrylate)-based resins are used. But authors should mention that other types of reactive resin can be used, and in that case, why methacrylate resins are preferred over different types of reactive systems.
Line 57: authors should specify whether the impact is negative or positive toward biological elements. In general, authors should be more specific to these types of printed material characteristics
Line 59: What does the author refer to as the degree of conversion, and why is this term introduced here and left without any explanation? Also, Why should the degree of conversion be important for biocompatible models?
Line 78: Even though some investigations have reported that in specific conditions and concentrations, BAPO can be used in medical applications (The same for TPO:):
10.1016/j.dental.2018.09.015
10.3390/nano10091788
10.3390/mi11030246
10.1016/j.dental.2018.09.015
Line 96: Authors should indicate the molecular weight of the resin used for their work.
Line 144: Authors should provide more information about the selection of this resin mixture. Why UDMA, Bis-EMA and TEGDMA were used in those percentages?
Line 126: This is a visible-light 3D printer. Authors should mention this from the beginning of their work. This is an important part of the work since the photoinitiators used to have different behavior upon UV or visible light.
Line 180: What L, a and b represent in the equation N°2?
Figure 5: The cell line under study should be better described in the figure caption.
Line 294: I suggest to the authors to describe what a*, and b* symbols represent? It is not clear what authors try to display in this experiment. Surely, a good description of the experiment can be found in ISO 4049:2019, anyhow authors should be explicit what they meant with these symbols
Line 365: more references are needed to better validate the statement above.
From lines 366 to 368: this part of the paragraph is confusing. It is unclear what the trade-off is that the authors are trying to explain.
Line 395: Why is color stability important for dental restoration and medical devices? It is unclear what authors are trying to justify in this part of the paragraph.
Line 396: Again, more explanation is missing. How can these properties affect the discoloration process? What are the mechanisms involved in the discoloration process, and how can they be avoided/reduced?
Line 429: Why does BAPO reveal a greater shrinkage than the other photoinitiators? This explanation is missing
Line 427: What are these factors? Any example?
Line 432: There are other reasons for over-polymerization in the XY plane. One is the time of printing irradiation (5.5 sec for 10o um of thickness), which can be adjusted to provide the proper dose to the resin. Another reason might be the printing thickness. These types of tests are missing. Authors should add more information about the kinetics of the reaction of their printable resin and how these can be tailored by changing the printing parameters (ex: irradiation time, thickness, photoinitiator concentration, light power (if possible). It should be interesting to find the optimum concentration of photoinitiators to obtain reliable printed structures.
Line 442: Authors should explain why the degree of conversions was similar for the three groups of photoinitiators? Why with TPO-L a higher degree of conversion is reached is TPO-L generates only two free radicals? Authors should also mention the figure/table where the information can be observed/read
Author Response
Reviewer #5 |
This paper titled: Biocompatible 3D printing resin for medical devices using ethyl (2,4,6-trimethylbenzoyl) phenylphosphinate as a photoinitiator presents a study using 3D printable resins using three different photoinitiators: BAPO, TPO and TPO-L. The objective is to show the better biocompatibility, color stability and printing accuracy of TPO-L compared to the other two types of photoinitiators. The TPO-L photoinitiator is a novel compound that can be efficiently used for 3D printing applications, however, in this manuscript. Some changes need to be made to increase the clarity of the investigation and scientific methodology. The overall English language is acceptable, with some changes throughout the paper. Sometimes, the sense of the sentences is unclear to understand. The beginning of the introduction is a bit dispersive, and the sense of the work might be lost. In general, the work is very interesting, and with a little effort, the work can be enhanced and easier to understand the methodology. |
||||||||||||||||||||||||
Comment |
Line 35: Which stages are the authors talking about? Can the authors provide a few examples to enrich this part? |
||||||||||||||||||||||||
Response to Comment |
Thank you for the valuable suggestions. The relevant information has been added according to your comments as follows: 1. Introduction ~ Conventional fabrication methods involve recording an impression of the treatment site, pouring a stone model and constructing a wax pattern. The wax pattern is invested and replaced with a permanent material such as metal, ceramic and acrylic resin. ~ |
||||||||||||||||||||||||
Comment |
Line 37: From a personal view, I find the CAM system a bit old-fashioned term. Since this paper is talking almost uniquely about 3D printing, I recommend authors use a more-updated term for referring to these technologies |
||||||||||||||||||||||||
Response to Comment |
According to your suggestion, we have removed the sentence as follows: 1. Introduction The CAM system is divided into additive manufacturing and subtractive manufacturing. |
||||||||||||||||||||||||
Comment |
Line 41: consuming less material compared to what? |
||||||||||||||||||||||||
Response to Comment |
According to your suggestion, we have added a relevant sentence. 1. Introduction A 3D printing system that corresponds to additive manufacturing has the advantage of being able to create complex shapes, personalized production and consuming less material than subtractive manufacturing [3,4]. |
||||||||||||||||||||||||
Comment |
Line 41: I suggest authors shift this part of the paragraph since 3D printable resins are introduced in the subsequent section. |
||||||||||||||||||||||||
Response to Comment |
According to your suggestion, line 41 has been moved to the subsequent section. 1. Introduction Dental 3D printing resins can be used to make prostheses such as surgical guides, crowns & bridges and dentures [7]. ~ |
||||||||||||||||||||||||
Comment |
Line 47: Authors should be careful of using indistinct methods and technology when discussing 3D printing. The same for light-curable and light-cured resins, there is a huge difference between these two terms. At last, DLP-based 3D printing methods also works with visible light and not only with UV. Indeed, as the same authors explain later, they use a visible-light 3D printer and not a UV-based one. |
||||||||||||||||||||||||
Response to Comment |
According to your suggestion, the terminology of the manuscript has been unified with light-cured resin and relevant information about the 3D printer has been added. 1. Introduction DLP technology is the method of irradiating light in tanks containing a light-cured resin that reacts with ultraviolet light or visible light and produces prosthetics through photopolymerization [3]. |
||||||||||||||||||||||||
Comment |
line 50: For the sake of printing resolution, authors should mention what each of the micromirror devices represents in terms of image resolution to connect to better explain why a "higher number of micromirrors, the higher is the resolution" |
||||||||||||||||||||||||
Response to Comment |
According to your suggestion, we have revised details as follows: 1. Introduction ~ When light irradiation, a digital micromirror device (DMD) is used to cure the entire layer of resin in the x-y axis at one time. The micromirrors, which act as light switches, project the light from the source as individual pixels onto the projection surface. Each micromirror represents one or more pixels in the projected image. The number of mirrors corresponds to the resolution of the projected image [5,6]. |
||||||||||||||||||||||||
Comment |
Line 53: Authors should provide some examples and/or references in this part of the paragraph. Line 54: what do the authors refer to matrix and fillers? Some example? Line 55: I suggest changing line 55. It is not clear what the purpose of the photoinitiator is: is it used to start the free radical polymerization, complete the polymerization, or both? Line 56: What authors refer to methacrylate-based material? A solid or liquid material? Why did the authors start to talk about methacrylate-based materials without being previously introduced? Typically, for light-based 3D printing, meth(acrylate)-based resins are used. But authors should mention that other types of reactive resin can be used, and in that case, why methacrylate resins are preferred over different types of reactive systems. Line 57: authors should specify whether the impact is negative or positive toward biological elements. In general, authors should be more specific to these types of printed material characteristics Line 59: What does the author refer to as the degree of conversion, and why is this term introduced here and left without any explanation? Also, Why should the degree of conversion be important for biocompatible models? |
||||||||||||||||||||||||
Response to Comment |
According to your suggestion, we have extensively revised some sections of the introduction part. 1. Introduction ~ The vast majority of dental light-cured resins are comprised of dimethacrylate resins [8,9]. Ingredients of light-cured resin are composed of a matrix, filler and photoinitiator to ensure the properties of the material [10]. For example, the resin matrix consists of dimethacrylate monomers such as BisGMA (bisphenol-glycidyl dimethacrylate), UDMA (urethane dimethacrylate), BisEMA (bisphenol-A-ethoxy dimethacrylate) or TEGDMA (triethyleneglycol dimethacrylate). The fillers were mainly used SiO2, TiO2, Al2O3 and ZrO2 nanoparticles [11]. A photoinitiator system is added to the resin matrix to trigger the process of free radicals for the polymerization reaction [12]. The degree of conversion (DC) can be represented as the extent to which monomers react to form polymers. The low degree of conversion has a negative impact on the biological, physical and mechanical properties of the polymer and ultimately determines the life of the restoration [13,14]. ~
8. Cramer, N.; Stansbury, J.; Bowman, C. Recent advances and developments in composite dental restorative materials. Journal of dental research 2011, 90, 402-416. 9. Cramer, N.B.; Couch, C.L.; Schreck, K.M.; Boulden, J.E.; Wydra, R.; Stansbury, J.W.; Bowman, C.N. Properties of methacrylate–thiol–ene formulations as dental restorative materials. Dental Materials 2010, 26, 799-806. 11. Wang, Y.; Zhu, M.; Zhu, X. Functional fillers for dental resin composites. Acta Biomaterialia 2021, 122, 50-65. |
||||||||||||||||||||||||
Comment |
Line 78: Even though some investigations have reported that in specific conditions and concentrations, BAPO can be used in medical applications (The same for TPO:): |
||||||||||||||||||||||||
Response to Comment |
According to your suggestion, we have revised details as follows: 1. Introduction ~ However, it is known to be more cytotoxic than TPO and CQ at the same concentration and induce discolouration [22]. ~ However, the higher cytotoxicity than same concentration of CQ and lower polymerization efficiency than BAPO remain problems [21]. |
||||||||||||||||||||||||
Comment |
Line 96: Authors should indicate the molecular weight of the resin used for their work. |
||||||||||||||||||||||||
Response to Comment |
According to your suggestion, we have added details as follows: 2. Materials and Methods 2.1. Materials ~ The molecular weight of each resin matrix was 376.4 g/mol for Bis-EMA, 470.56 g/mol for UDMA, and 286.32 g/mol for TEGDMA. ~ |
||||||||||||||||||||||||
Comment |
Line 144: Authors should provide more information about the selection of this resin mixture. Why UDMA, Bis-EMA and TEGDMA were used in those percentages? |
||||||||||||||||||||||||
Response to Comment |
According to your suggestion, we have revised the sentence and added references. 2. Materials and Methods 2.3. Preparation of 3D printing resin matrix The experimental 3D printing resin matrix was formulated using 70 wt% UDMA, 20 wt% Bis-EMA and 10 wt% TEGDMA with a high degree of conversion according to references [28]. ~ |
||||||||||||||||||||||||
Comment |
Line 126: This is a visible-light 3D printer. Authors should mention this from the beginning of their work. This is an important part of the work since the photoinitiators used to have different behavior upon UV or visible light. |
||||||||||||||||||||||||
Response to Comment |
According to your suggestion, we have revised details as follows: 1. Introduction ~ DLP technology is the method of irradiating light in tanks containing a light-cured resin that reacts with ultraviolet light or visible light and produces prosthetics through photopolymerization [3]. 2. Materials and Methods 2.4. Preparation of 3D printed specimens ~ All specimens were manufactured using a DLP printer (D2; Veltz 3D Co., Incheon, Korea). The wavelength was 405 nm corresponding to visible light, and the accuracy was ± 2 µm. ~ |
||||||||||||||||||||||||
Comment |
Line 180: What L, a and b represent in the equation N°2? Line 294: I suggest to the authors to describe what a*, and b* symbols represent? It is not clear what authors try to display in this experiment. Surely, a good description of the experiment can be found in ISO 4049:2019, anyhow authors should be explicit what they meant with these symbols |
||||||||||||||||||||||||
Response to Comment |
According to your suggestion, the relevant sentence has been added to the materials and methods part to clearly explain the L*, a*, and b* symbol. 2. Materials and Methods 2.6. Colour stability ~ In the CIELAB system, the location of a particular shade in the colour space is defined by three coordinates: L*, a*, and b*. L* describes the lightness of the object being evaluated. The a* value defines the colour on the red-green axis and b* on the yellow-blue axis. ~ |
||||||||||||||||||||||||
Comment |
Figure 5: The cell line under study should be better described in the figure caption. |
||||||||||||||||||||||||
Response to Comment |
According to your suggestions, we added cell lines to the sentence and converted the data into a table format to make it easier to understand from the reader. 3. Results 3.2. Cytotoxicity test The results of the cytotoxicity test of 3D printed resin using the L‐929 mouse fibroblasts are shown in Figure 6. ~ The results of the cytotoxicity test of each photoinitiator using the L‐929 cell line are shown in Table 1. ~
Figure 5. Cell viability of 3D printed resin groups. Differences in lowercase alphabetical letters above the bar graph indicate significant differences of each group (p<0.05). Table 1. Cell viability of each photoinitiator (%).
The same lowercase letter in the same column indicates no significant difference (p>0.05). |
||||||||||||||||||||||||
Comment |
Line 365: more references are needed to better validate the statement above. From lines 366 to 368: this part of the paragraph is confusing. It is unclear what the trade-off is that the authors are trying to explain. |
||||||||||||||||||||||||
Response to Comment |
According to your suggestion, we have added references and revised the sentence as follows: 4. Discussion ~The selection of an appropriate photoinitiator is essential to obtain the properties of the desired polymer [9,11,17]. A suitable photoinitiator should correlate absorption characteristics of the photoinitiator and emission characteristics of the light source. In addition, it should be non-cytotoxic and not cause discolouration [22,35].~ 9. Meereis, C.T.; Leal, F.B.; Lima, G.S.; de Carvalho, R.V.; Piva, E.; Ogliari, F.A. Bapo as an alternative photoinitiator for the radical polymerization of dental resins. Dental materials 2014, 30, 945-953.
11. Cramer, N.B.; Couch, C.L.; Schreck, K.M.; Boulden, J.E.; Wydra, R.; Stansbury, J.W.; Bowman, C.N. Properties of methacrylate–thiol–ene formulations as dental restorative materials. Dental Materials 2010, 26, 799-806.
17. Wang, Y.; Zhu, M.; Zhu, X. Functional fillers for dental resin composites. Acta Biomaterialia 2021, 122, 50-65.
22. Zeng, B.; Cai, Z.; Lalevée, J.; Yang, Q.; Lai, H.; Xiao, P.; Liu, J.; Xing, F. Cytotoxic and cytocompatible comparison among seven photoinitiators-triggered polymers in different tissue cells. Toxicology in Vitro 2021, 72, 105103.
35. Tomal, W.; Ortyl, J. Water-soluble photoinitiators in biomedical applications. Polymers 2020, 12, 1073.
|
||||||||||||||||||||||||
Comment |
Line 395: Why is color stability important for dental restoration and medical devices? It is unclear what authors are trying to justify in this part of the paragraph. Line 396: Again, more explanation is missing. How can these properties affect the discoloration process? What are the mechanisms involved in the discoloration process, and how can they be avoided/reduced? |
||||||||||||||||||||||||
Response to Comment |
According to your suggestion, we have revised some sections of the discussion part. 4. Discussion ~ Esthetics is very important for teeth. For long-term restorations in the oral cavity, the color of the resin prostheses has a significant impact on patient satisfaction. Therefore, Colour stability is an important factor in the success and longevity of restorations [37]. The discolouration of resins can be influenced by a number of factors such as incomplete polymerization, water sorption, chemical reactivity, diet, oral hygiene or surface roughness of the restoration [38]. The intrinsic discoloration among the various causes of discoloration mainly depends on the initiator system used in the resins as well as on the degree of polymerization [39].~ The photoinitiator with a concentration that does not generate the remaining radicals could be expected to avoid or reduce discoloration. |
||||||||||||||||||||||||
Comment |
Line 427: What are these factors? Any example? |
||||||||||||||||||||||||
Response to Comment |
According to your suggestion, we have revised details as follows: 4. Discussion ~ Many factors affect the shrinkage in dental resin. These factors can be separated into material formulation factors (filler content, monomer structure, filler/matrix interactions, additives, etc.) and material polymerization factors (polymerization rate, i.e. catalyst and inhibitor concentration, external constraint conditions, curing method, etc.) [45]. Photoinitiators affected shrinkage as material polymerization factors. ~ |
||||||||||||||||||||||||
Comment |
Line 429: Why does BAPO reveal a greater shrinkage than the other photoinitiators? This explanation is missing Line 432: There are other reasons for over-polymerization in the XY plane. One is the time of printing irradiation (5.5 sec for 100 um of thickness), which can be adjusted to provide the proper dose to the resin. Another reason might be the printing thickness. These types of tests are missing. Authors should add more information about the kinetics of the reaction of their printable resin and how these can be tailored by changing the printing parameters (ex: irradiation time, thickness, photoinitiator concentration, light power (if possible). It should be interesting to find the optimum concentration of photoinitiators to obtain reliable printed structures. |
||||||||||||||||||||||||
Response to Comment |
With your comment, we have clearly identified the limitations of the current study. We have added limitations and the need for further research to the discussion part. 4. Discussion ~ However, current research is limited because there were no experiments with various 3D printing parameters for shrinkage and over-polymerization. In future research, it would be necessary to change parameters such as irradiation time, layer thickness and photoinitiator concentration and study the optimum concentration of photoinitiators and resin shrinkage rate. |
||||||||||||||||||||||||
Comment |
Line 442: Authors should explain why the degree of conversions was similar for the three groups of photoinitiators? Why with TPO-L a higher degree of conversion is reached is TPO-L generates only two free radicals? Authors should also mention the figure/table where the information can be observed/read |
||||||||||||||||||||||||
Response to Comment |
According to your suggestion, we modified the discussion part by adding FT-IR spectra and references that measured the curing kinetics of photoinitiators under the same molar concentration. 2. Materials and Methods 2.8. Degree of conversion Figure 3. Comparison of before and after conversion of FT-IR spectra. (A) Comparison of the BAPO group, (B) comparison of the TPO group, (C) comparison of the TPO-L group. The black dashed line shows the after conversion. 4. Discussion ~ Figure 3 shows that the FT-IR spectra of each group are similar. Therefore, it could be expected that the degree of conversion of the experimental groups will be similar. ~ According to Macarie et al., conversion results of BAPO group could be explained by the competition between the reactions of primary radicals and double bonds during the initiation stage due to the higher concentration of radicals [48]. In addition, Lebedevaiteet al. confirmed the effect of photoinitiators on photocrosslinking kinetics by comparison of the G’ curves of the resins. 3 mol% BAPO demonstrated higher photocrosslinking kinetics compared to photoinitiators of the same concentration. However the G' plateau is similar to other photoinitiators [49]. That explains why the degree of conversion in the experimental groups is not significantly different. 48. Macarie, L.; Ilia, G. The influence of temperature and photoinitiator concentration on photoinitiated polymerization of diacrylate monomer. Central European Journal of Chemistry 2005, 3, 721-730. 49. Lebedevaite, M.; Ostrauskaite, J. Influence of photoinitiator and temperature on photocross-linking kinetics of acrylated epoxidized soybean oil and properties of the resulting polymers. Industrial Crops and Products 2021, 161, 113210. |
Round 2
Reviewer 2 Report
The authors have performed a minimal review. The authors have made no changes to any of the fundamental aspects. This manuscript does not allow for any solid conclusions. The authors' discussion has not been modified with the indications. This manuscript must be rejected.
Author Response
Now the manuscript has been substantially modified with modifications highlighted with track changes (including discussions).
Reviewer 3 Report
It can be accepted now.
Author Response
Thank you for your comment.
Round 3
Reviewer 2 Report
The authors have not made any substantial changes. This is a manuscript with very substantial doubts in the methodology and results. The quality of the results is questionable. The authors present an imprecise discussion. The results are not supported by indirect and categorical conclusions. The manuscript has numerous errors.
Author Response
Sincere apologies of not meeting your expectation despite the revision. The purpose of this manuscript was to investigate possibilities of using TPO-L as phothoinitiator for 3D printing medical devices, which may have improved biocompatibility (lower cytotoxicity) while maintaining other features as 3D printing resins. Comparison was made with previously used photointiators; TPO and BAPO.
The main results are therefore cytotoxicity (cell viability), color changes, and dimensional accuracy that related to biocompatibility and polymerization of resins.
These points are now more clearly stated in Introduction and Dicussion, so that results are now linked with conlcusions.
Methodology and results are now supported by references and reviewed by 4 other reviewers with 3 rounds of reviews. Limitations are stated as we understand that the research shows only the findings related to above purposes.
Finally, we have now carried out English language editing provided by MDPI, where certificate is now attached.
Hopefully, this would have reduced errors and resolved issues in your kind comments.
